



# Singular Vector Decomposition (SVD) of satellite datasets: relation between cloud properties and climate indices

Elisa Carboni[1,2,*], Gareth E. Thomas[1,2,*], Richard Siddans[1,2], and Brian Kerridge[1,2]

[1]RAL Space, Science and Technology Facilities Council, Rutherford Appleton Laboratory, Harwell Campus, Didcot, UK
[2]National Centre for Earth Observation, Rutherford Appleton Laboratory, Harwell Campus, Didcot, UK
[*]These authors contributed equally to this work.

**Correspondence:** Elisa Carboni (elisa.carboni@stfc.ac.uk)

**Abstract.** We describe a technique using singular vector decomposition (SVD), that can identify the spatial patterns that best describe the temporal variability of a global satellite dataset. These patterns, and their temporal evolution, are then correlated with established climate indices. We apply this technique to datasets of cloud properties over three decades, derived from five visible/IR imagers ((A)ATSR, SLSTR-A/-B and MODIS and jointly from the IR and microwave sounders on MetOp (IASI, MHS,AMSU-A), but it can be more generically used to extract the pattern of variability of any regular gridded dataset such as different parameters from satellite products and models. The leading singular vector for these three independent global data sets, on both cloud fraction and cloud-top height, from these polar orbiting satellites covering different time periods, is found to be strongly correlated with the ENSO index. The SVD approach could potentially offer a new tool for using global satellite observations in assessing global climate model (GCM) performance.

## 1 Introduction

Change in cloud properties is the main atmospheric contributor to change in the Earth's radiative balance. Clouds reflect solar shortwave radiation and absorb terrestrial thermal radiation and, depending on location, altitude and optical properties, they can have warming or cooling effects. Thus, change in cloud properties is an important feedback mechanism in climate.

The interdependence of cloud properties and surface temperature is well established (Andrews and Webb, 2018; Ceppi and Fueglistaler, 2021) and broadly intuitive. The surface temperature is a primary driver of atmospheric convection, which in turn is a dominant factor in cloud formation and properties. Conversely, surface temperature is largely driven by the radiative balance of the overlying atmosphere (especially over the land), which is dominated by the presence and radiative properties of overlying clouds. Ceppi and Fueglistaler (2021) show that in the tropics, change in the spatial distribution of surface temperature affects the temperature at higher levels in the atmosphere, which in turn changes the amount of low cloud and thereby the radiation budget. Ceppi and Nowack (2021) find that global cloud feedback is positive and dominated by the sensitivity of clouds to surface temperature and tropospheric stability. However accurately describing and modelling the scale, spatial and temporal variability of this feedback is not a simple task.

Furthermore, the link between global cloud properties and the well known El Niño Southern Oscillation (ENSO) climate-index is also been widely observed . ENSO quantifies the irregular oscillation of the climate system between two semi-stable


states characterised by sea-surface temperatures (SSTs) in the tropical eastern Pacific that are either warmer than average, a
    state known as El Niño, or cooler, known as La Niña. The ENSO phenomenon is the largest source of global SST variability
    on inter-annual time scales. A multitude of studies have presented analysis linking cloud variations with ENSO.

    In 150 years of pre-industrial climate simulations, Yang et al. (2016) show that ENSO was strongly correlated with changes
    in the cloud radiative effect, linked to changes in the fraction of cloud cover. Davies (2019) used 18 years of cloud-top height
anomalies to find teleconnection patterns between the El Niño and La Niña phases, while Madenach et al. (2019) used 14 years
    of MODIS data to demonstrate that the variability of cloud-top height is highly influenced by ENSO and associated large-scale
    atmospheric dynamics. (Li et al., 2021) use the Empirical Orthogonal Function analysis with CloudSat data of total cloud cover
    and find that the principal oscillation mode is associated with ENSO.

    However, the relationship between cloud variability and ENSO is complex and can vary depending on the specific type of
cloud and the phase of the ENSO cycle. Further research is needed to fully understand this relationship and its implications for
    climate change.

    In this paper we describe a technique that can identify the spatial patterns that best describe the temporal variability of a
    gridded, global satellite dataset. These patterns, and the temporal evolution of the weights associated with them, can then be
    correlated with other, independent global data or established climate indices. This approach can provide insights into the un-
derlying causes of observed changes in a particular dataset and potentially offers a new tool to use global satellite observations
    to compare with and assess global climate model (GCM) performance. We apply this technique to datasets of cloud properties,
    but it can be more generically used to extract the pattern of variability of any regular gridded dataset.

    We apply a singular value decomposition (SVD) analysis to the de-seasonalised time-series of maps of monthly-mean
    cloud properties from three separate global satellite products. These are: (i) the Cloud_cci and Copernicus Climate Change
Service (C3S) data record derived from the Along-Track Scanning Radiometer (ATSR) and Sea and Land Surface Temperature
    Radiometer (SLSTR) series of instruments; (ii) the Moderate Resolution Imaging Spectroradiometer (MODIS) Collection
    6.1 dataset (from the Terra satellite), and (iii) the Infrared/Microwave Sounder (IMS) retrieval scheme applied to combined
    measurements from the Infrared Atmospheric Sounding Interferometer (IASI), Advanced Microwave Sounding Unit (AMSU)
    and Microwave Humidity Sounder (MHS). This analysis produces maps of mathematically orthogonal modes of variability
(the singular vectors, SV) and weights which define the contribution of each singular vector to the total and their variability
    through time.

    The singular vector maps produced by this process can be qualitatively compared to maps of variability associated with
    different parameters, such as the SST patterns associated with ENSO, and in instances where a singular vector resembles a
    particular climate index, the timeseries of weights can be compared to the climate index itself. This is useful for two main
reasons:

1. It gives a clear indication as to which modes of variability are dominant in global cloud fields.

2. It provides a mechanism to use global observations of cloud properties (and, indeed any other global satellite product)
   to access other datasets which are not directly comparable.





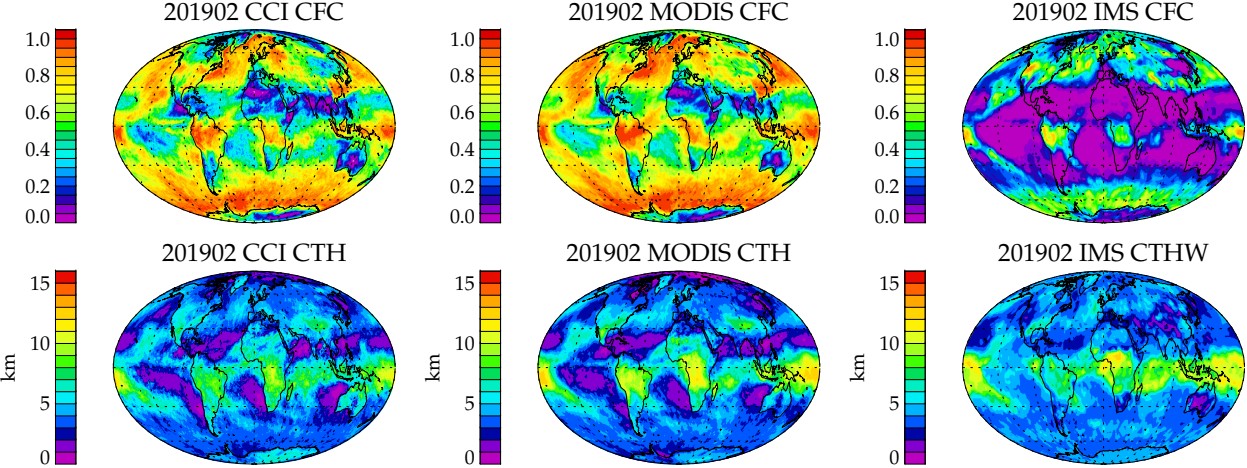

**Figure 1.** Cloud_cci/C3S, MODIS and IMS exemple of monthly maps for the month February 2019. The first row shows the CFC, second row shows CTH.

Section 2 describes the cloud properties datasets used, while section 3 describes the methodology and section 4 shows the
results obtained for cloud fraction and cloud top high and the relation with climate indices such as the El Nino Southern Oscillation (ENSO).

## 2 Datasets

Figure 1 shows the monthly mean cloud fraction (CFC) and cloud top height (CTH) for the three datasets considered (Cloud_cci/C3S, MODIS and IMS) for the month of February 2019. All three datasets have global coverage and show similar spatial patterns
of CFC and CTH, but there is some disagreement in the magnitude. The IMS data in particular presents lower CFC and higher CTH compared to the other two datasets. This can be explained by the fact that IMS cloud retrieval uses thermal infrared information only, at a spatial resolution of 25 km, while Cloud_cci and MODIS use visible and infrared channels together, at a spatial resolution of approximately 1 km. Thus IMS can be expected to have less sensitivity to small clouds, due to its coarse sampling, less sensitivity for cloud close to the surface, due to a lack of thermal contrast, but be more sensitive to optically
thin, high altitude clouds, due to its high spectral sampling.

### 2.1 Cloud CCI

We use version 3 of the Cloud_cci Along-Track Scanning Radiometer (ATSR) and Advanced ATSR (AATSR) data set (Poulsen et al., 2019). The data set was created for the European Space Agency (ESA) Cloud_cci (Climate Change Initiative) programme. The cloud properties were retrieved from the second ATSR (ATSR-2) on board the second European
Remote Sensing Satellite (ERS-2) spanning 1995–2003, and the AATSR on board Envisat, which spanned 2002–2012. To-





gether, these two instruments are commonly referred to as (A)ATSR. The data are comprised of a comprehensive set of cloud properties: cloud top height, temperature, pressure, spectral albedo, cloud effective emissivity, effective radius and optical thickness, alongside derived liquid and ice water path. Each retrieval is provided with its associated uncertainty. The cloud properties are derived using the Optimal Retrieval of Aerosol and Cloud (ORAC) retrieval scheme (McGarragh et al., 2018; Sus et al., 2018) and are accompanied by high-resolution top- and bottom-of-atmosphere short- and longwave fluxes that have been derived from the retrieved cloud properties using the Bugsrad radiative transfer model (Stephens et al., 2001) (see https://biocycle.atmos.colostate.edu/shiny/BUGSrad/). Together the full processing scheme is known as the Community Cloud retrieval for Climate (CC4CL).

The (A)ATSR Cloud_cci product has also been brokered to the Copernicus Climate Change Service (C3S), providing Climate Data Records (CDRs) on cloud properties[1], surface radiation budget[2] and Earth radiation budget[3]. Under C3S the (A)ATSR CDR has been supplemented by an Interim-CDR (ICDR) derived from the Sea and Land Surface Temperature Radiometer (SLSTR), on board the Copernicus Sentinel-3 satellites. SLSTR is an operational follow-on to the ATSR series of instruments and the ICDR has been derived using the same CC4CL processing chain as the Cloud_cci (A)ATSR product. The ICDR begins in January 2017, with data from Sentinel-3A, with data from Sentinel-3B being included from October 2018 onward.

The data analysed here consist of Level-3C monthly-mean composites of cloud properties on a regular $0.1 \times 0.1°$ latitude-longitude grid.

## 2.2  MODIS

We used CFC and CTH from the Moderate Resolution Imaging Spectroradiometer (MODIS) Collection 6.1 Terra dataset[4]. The MODIS-Terra product was selected over the Aqua product, as Terra has a similar overpass time to (A)ATSR and SLSTR (with a descending equatorial crossing at 10:30 local solar time). MODIS-Terra views the entire Earth's surface every 1 to 2 days, acquiring data in 36 spectral bands, with observations starting in 2000. We have used the level-3 MODIS gridded atmosphere monthly global products - MOD08_M3. They contain monthly $1° \times 1°$ degree grid average values of atmospheric parameters related to atmospheric aerosol particle properties, total ozone burden, atmospheric water vapour, cloud optical and physical properties, and atmospheric stability indices.

## 2.3  IMS

The RAL (Rutherford Appleton Laboratory) Infra-red/Microwave Sounder (IMS) retrieval scheme (Siddans, 2019) uses an optimal estimation (OE) spectral fitting procedure to retrieve atmospheric and surface parameters jointly from co-located measurements by the sounders IASI (Infrared Atmospheric Sounding Interferometer), AMSU (Advanced Microwave Sounding Unit) and MHS (Microwave Humidity Sounder) on the MetOp spacecraft series. The MetOp satellites are sun-synchronous

---

[1]Cloud properties CDR: https://doi.org/10.24381/cds.68653055

[2]Surface radiation CDR: https://doi.org/10.24381/cds.cea58b5a

[3]Earth radiation CDR: https://doi.org/10.24381/cds.85a8f66e

[4]MODIS Terra Collection 6.1 cloud product: https://dx.doi.org/10.5067/MODIS/MOD08_M3.061



operational weather satellites, with an equatorial crossing at 9:30 local solar time, which have been supplying data since mid-2007. IMS uses RTTOV 12 (Radiative Transfer for TOVS) (Hocking, 2018) as the forward radiative transfer model. IMS retrieves profiles of temperature, humidity and ozone together with total column amounts of a number of minor trace gases, surface emissivity cloud and aerosol parameters. In the version of the IMS data used here, cloud is modelled as a

scattering layer with profile shape defined to be Gaussian in mass mixing ratio units with a full-width-half-maximum of 2km. Retrieved parameters are cloud optical depth, effective radius and the (centre) height of the Gaussian mass-mixing ratio profile. Differences between observed brightness temperatures around 11.07, 12.02, 9.32, 8.96 and 8.3 microns are used to determine if a optical properties for ice or liquid cloud are used. In case ice cloud is selected then IMS runs RTTOV with the "SSEC/Baum" cloud optical properties; in case liquid cloud is selected then the "Deff" properties are assumed (see the RTTOV12 user guide

(Hocking, 2018)). Cloud is only explicitly modelled in the simulation of thermal infra-red (IASI) radiances; the impact of cloud on microwave observations is partially accommodated by the co-retrieved microwave surface emissivity. Scenes where this approach is not sufficient to simultaneously explain the observations in all three instruments yield poor fit residuals and are consequently excluded from this analysis.

       IMS does not retrieve cloud fraction as such, only a scene effective cloud optical depth, $\tau_{eff}$ (expected to be negligibly small

in cloud-free scenes). Here we use an effective cloud fraction, $f_{eff}$ to compare to the other cloud datasets:

$$f_{eff} = 1 - \exp\left(-\tau_{eff}\right) \tag{1}$$

The monthly mean L3 IMS cloud height is computed weighted by the effective cloud fraction (such that the monthly mean cloud height is dominated by the height in scenes with relatively high optical depth/effective fraction) on $1° \times 1°$ degree grid.

## 2.4    Climate indices

We considered the monthly atmospheric and ocean time series climate indices[5], obtained from the USA National Oceanic and Atmospheric Admistration (NOAA). This list includes teleconnections, atmosphere, precipitation, ENSO and surface temperature indices. The complete list is reported in table 1.

## 3    Method

The analysis described in this session is, applied to two cloud parameters: cloud fractional cover (CFC) and cloud top height (CTH), over a latitude range of 60° S and 60° N. Although all three datasets provide coverage to latitudes well above 80°, both the MODIS and cloud_cci products are known to have limitations over ice and snow surfaces, so this analysis has been limited to mid- and tropical-latitudes. The same methodology can be applied to any time series of a parameter that is presented in the form of a map, either from satellite observations or models, of cloud properties, radiative fluxes or other geophysical

[5]NOAA climate indices database: https://psl.noaa.gov/data/climateindices/



**Table 1.** Climate indices, from NOAA, used in the fit of the SVD temporal weights

| | |
|---|---|
| AAO | Antarctic Oscillation |
| AMM | Atlantic Meridional Mode |
| AMO | Atlantic Multi-decadal oscillation |
| AMOsm | AMO (smoothed) |
| AO | Arctic Oscillation |
| AtlanticTripole | Atlantic Tripole SST EOF |
| BEST | Bivariate ENSO Timeseries |
| BESTlng | Bivariate ENSO Timeseries (long) |
| Brazil | Northeast Brazil Rainfall Anomaly |
| CAR | Caribbean SST Index |
| DMI | Indian Ocean Dipole (IOD) |
| | aka Dipole Mode Index (DMI) |
| EA | Eastern Atlantic/Western Russia |
| ENSO | El Nino Southern Oscillation |
| EPNP | East Pacific/North Pacific Oscillation |
| ESPI | ENSO Precipitation Index |
| Glaam | Globally Integrated Angular Momentum |
| GMSST | Global Mean Land/Ocean Temperature Index |
| Hurricane | Monthly Totals Atlantic Hurricanes and Storms |
| India | Central Indian Precipitation |
| MEI V2 | Multivariate ENSO Index V2 |
| NAOJONES | North Atlantic Oscillation (Jones) |
| NAO | North Atlantic Oscillation |
| Nina12 | Extreme Eastern Tropical Pacific SST |
| Nina34 | East Central Tropical Pacific SST |
| Nina3 | Eastern Tropical Pacific SST |
| Nina4 | Central Tropical Pacific SST |
| NOI | Northern Oscillation Index |
| NP | North Pacific Pattern |
| NTA | North Tropical Atlantic Index |
| ONI | Oceanic Nino Index |
| PacificWarm | Pacific Warmpool Area Average |
| PDO | Pacific Decadal Oscillation |
| PNA | Pacific North American Index |
| QBO | Quasi-Biennial Oscillation |
| Sahel | Sahel Standardized Rainfall |
| SOI | Southern Oscillation Index |
| Solar | Solar Flux (10.7cm) |
| TNA | Tropical Northern Atlantic Index |
| TNI | Trans-Nino index of El Nino Evolution |
| TPI | Tripole Index |
| | for the Interdecadal Pacific Oscillation |
| TropicalEOF | 1st EOF of Tropical Pacific SST |
| TSA | Tropical Southern Atlantic Index |
| WHWP | Western Hemisphere Warm Pool |
| WP | Western Pacific Index |



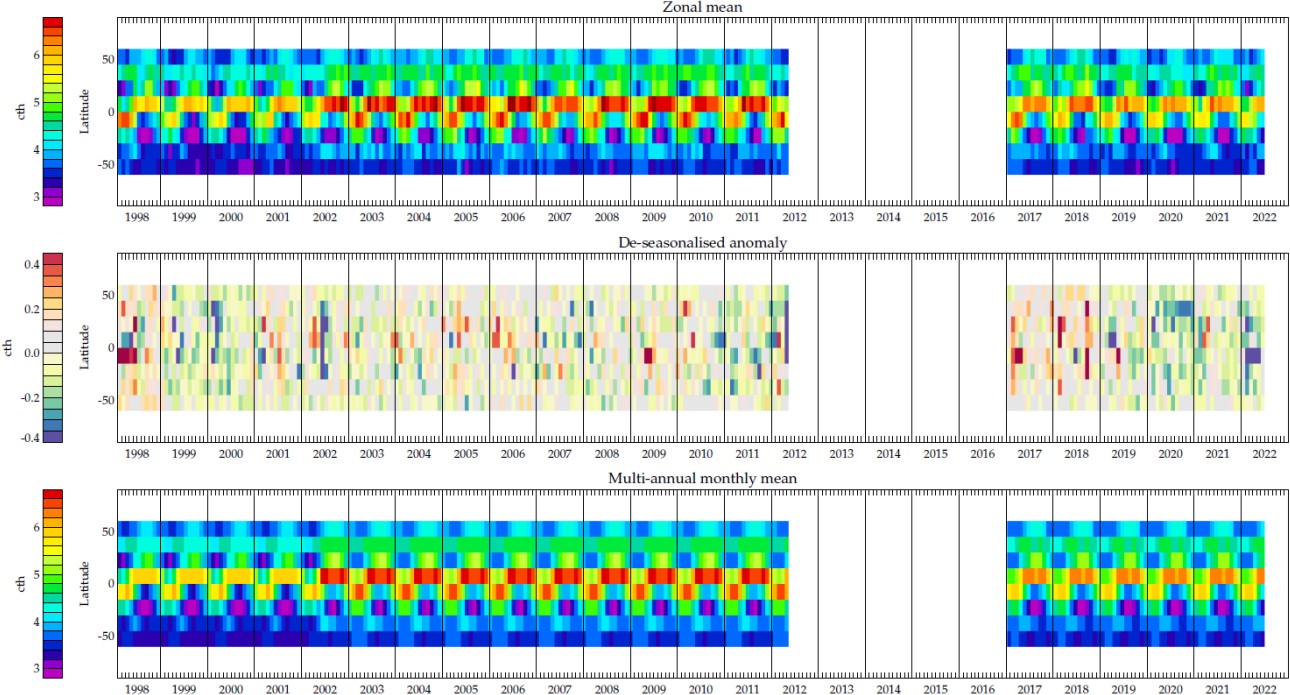

**Figure 2.** This figure shows from top to bottom: Zonal mean of CTH as function of time for the cloud_cci dataset, zonal mean of the de-sesonalized anomaly of CTH, the multi annual momthly mean (or annual cycle/sesonal contribution) of CTH.

parameters. Here we work with monthly mean (Level-3C), data but the method can be used with different time scales, as long as the temporal sampling is consistent across all datasets compared.

### 3.1 De-seasonalised anomaly

We start with a monthly L3 cloud property dataset. In all cases this is defined on a regularly spaced longitude-latitude grid. These data a converted into a de-seasonalised anomaly by subtracting the annual cycle (or seasonal contribution) from each monthly mean. I.e. for each L3 grid cell in each month, we subtract the mean, over all years, of the values in the same grid cell for that calendar month. Note that this is carried out separately for the data from each instrument included in the analysis, such that time-invariant instrumental/retrieval bias is subtracted from the anomaly, together with the annual cycle. We call the result the de-seasonalised anomaly data or more simply 'anomaly'.

The same approach is adopted to compute de-seasonalised anomalies of the climate indices before determining their correlation with the time-series of the cloud data (see section 3.4, below).

Figure 2 shows the the zonal average time series (for the paramenter CTH of the Cloud_cci dataset) and the corresponding zonal average of the de-sesonalized anomaly, together with the instrument dependent annual cycle.





## 3.2 Singular vector decomposition (SVD)

SVD is a mathematical technique used to decompose a matrix of numerical data into a set of orthogonal singular vectors and weights (also know as singular values), which can help identify patterns and trends in the data. 3D fields of gridded cloud properties are stored in a matrix, $\mathbf{A}$, with $n$ rows corresponding to L3 grid points[6] and $m$ monthly time steps. In the current case, $m < n$.

The SVD, implemented using the standard routine `LA_SVD` from LAPACK (Anderson et al., 1999), decomposes the matrix $\mathbf{A}$ such that

$$\mathbf{A} = \mathbf{USV}^T \tag{2}$$

Where $\mathbf{S}$ is a diagonal square matrix of dimension $m$. The diagonal elements are all positive and contain the singular values of $\mathbf{A}$. $\mathbf{V}$ is a matrix with $n$ rows each containing one of the $m$ orthogonal *spatial singular vectors*, defined such that $\mathbf{VV}^T = \mathbf{I}$, the unit matrix of dimension $m$. These spatial singular vectors (SV) can be reformed as a maps describing orthogonal modes of spatial variability. $\mathbf{U}$ is a square matrix of dimension $m$, defined such that $\mathbf{UU}^T = \mathbf{I}$, which can be understood as containing weights (normalised by the singular values in $\mathbf{S}$) describing the temporal evolution of each of the spatial singular vectors. The variance of the anomaly over time of each of the spatial singular vectors can be derived from the corresponding singular value in $\mathbf{S}$.

The matrices are organised such that $\mathbf{S}$ contains singular values in descending order so that the first row of $\mathbf{V}$ contains the spatial singular vector associated with largest variance over time, i.e. the *leading* singular vector or dominant mode of variance.

## 3.3 Normalisation of the spatial singular vectors

The spatial singular vectors in $\mathbf{V}$ and their corresponding temporal weights in $\mathbf{U}$ are normalised such that the sum of their squared values is 1. The singular values in $\mathbf{S}$ have the units of the cloud property (e.g. cloud height in km). In order to present results in a manner which is simpler to relate to localised perturbations in the cloud property, as opposed to standard deviation over the L3 grid, we adopt the following approach.

1. The 99th percentile of the absolute values in each spatial singular vector is determined, defining a scalar value, $v_i^{99}$, for each singular vector denoted by index $i$. For presentation purposes (e.g. in figures 3 to 5), the spatial singular vector is normalised by $v_i^{99}$, such that at least 99% of its values fall in the range +/-1. The 99th percentile is used here, rather than the maximum value to avoid artefacts from noise or outliers obscuring the presentation of the results.

2. The corresponding temporal coefficients, $U_{ij}$, where $j$ is temporal index, are presented as $U'_{ij} = S_i v_i^{99} U_{ij}$. This means that the time-series of the $U'_i j$ are in the units of the cloud physical property and the values correspond to the 99th

---

[6]The organisation of the grid point values in this matrix can be arbitrary, but we store the values in order of increasing longitude first then increasing latitude. Resulting singular vectors retain this organisation so can be simply transformed to 2D maps.





percentile local perturbation associated with that mode of variability. (The location of this local "maximum" perturbation is clear from the presentation of the normalised spatial singular vector.)

Note that the sign of all values in $\mathbf{U}$ and $\mathbf{V}$ can be simultaneously reversed while preserving the validity of 2. Again to present results in a more intuitive manner, we consistently adjust the sign of all elements in $\mathbf{U}$ and $\mathbf{V}$ such that the strongest correlation of the leading singular vector with any of the climate indices is ensured to be positive (see section 3.4). There is no further constraint on the sign of correlations with any of the remaining singular vectors or climate indices.

### 3.4 Fit with the climate indices

The climate indices (section 2.4) are fitted to the time-series of temporal weights $U'_{ij}$ as follows:

1. The climate indices (defined per month) are sampled to the $m$ time steps of the cloud data. For each climate index, a series off time-lags are considered: $-6 \leq l \leq 6$ months in one month steps. The de-seasonalised anomaly of each lagged climate index is computed as described in 3.1. These are stored in $L_{k(j-l)}$ where index $k$ denotes a specific climate index from table 1.

2. The correlation of each climate index with the $U'_{ij}$ is computed:

$$C_{ikl} = \frac{\langle(U'_{ij} - \langle U'_{ij}\rangle)(L_{k(j-l)} - \langle L_{k(j-l)}\rangle)\rangle}{\sqrt{\langle(U'_{ij} - \langle U'_{ij}\rangle)^2\rangle\langle(L_{i(j-l)} - \langle L_{i(j-l)}\rangle)^2\rangle}} \tag{3}$$

   where $\langle.\rangle$ denotes expectation value over index $j$, i.e. the mean over all time steps of the quantity inside the angle brackets.

3. The climate index $k$ with the maximum absolute value of $C_{ikl}$, for any lag, is identified.

4. A linear least-squares fit procedure is applied to determine the coefficients $a_{i0}$ (offset), $a_{i1}$ (linear trend), $a_{i2}$ (scaling for the de-seasonalised, lagged climate index) in the following:

$$U'_{ij} = a_{i0} + a_{i1}j + a_{i2}(L_{kj} - \langle L_{k(j-l)}\rangle) + \Delta_{ij} \tag{4}$$

   The fit is performed to minimise the mean squared fit residual $\langle\Delta^2_{ij}\rangle$. In the fit, the climate index lag, $l$, which gave the maximum correlation is used.

This procedure is repeated for each singular vector (index $i$).

## 4 Results and discussion

To illustrate the approach, figure 3 shows, in the left-hand column, the first four normalised spatial singular vectors (SV) for cloud top height (CTH) derived from the cloud_cci data. Panels on the right show the temporal weights associated with the





singular vectors, as a function of the months (in units of km "maximum" localised perturbation), together with the fits to the climate indices for which the temporal correlation is largest in each case. Here the first time series of temporal weights is very well reconstructed by fitting with the BEST (Bivariate ENSO Timeseries) index, yielding a correlation coefficient of 0.87.

The index which best fits the second singular vector time series is Nina12 (another ENSO index), however the correlation coefficient has dropped to 0.57 and for subsequent SVs correlations with best fitting climate indices drop progressively and are down to less then 0.2 after the fourth SV. The subsequent SVs are not easily associated with climate indices, exhibiting much lower correlations and more 'noisy' spatial structure in their maps.

Figure 4 shows the results for the leading SV for the three satellite datasets for CTH, and fig. 5 shows the results for CFC.
Though not identical, the SV maps of these three datasets show very similar patterns, with high CTH in the west pacific and low CTH over Indonesia, in correspondence with a positive ENSO index. The three correlation coefficients with the BEST climate index are higher than 0.86.

It is important to note that the magnitude of the temporal weighting has not been normalized and nevertheless they have the same magnitude in the datasets considered, for example reaching values of 3-4 km in correspondence with the ENSO time-
series peaks (2009/2010 and 2015/2016). This shows that the magnitudes of the CTH anomaly, represented by the first SV, are consistent between the three satellite datasets.

Similarly, for CFC, there is a high correlation (>0.81) between temporal weights and the BEST climate index. Furthermore, cloud fraction anomaly, represented by the first singular vector, is attributable to ENSO and can reach values of 20%. In particular CFC decreases in the area over Indonesia during El Nino events and this decrease of cloud fraction is consistent
between the three datasets within few percent.

The first SV, of all considered cloud properties datasets, show the same spatial pattern: strong positive CFC and CTH anomalies in the central tropical Pacific region coupled with strong negative CFC and CTH anomalies in the Indonesian region. The fitting of the temporal weights of the first SV with climate indices show that this behavior is in close correspondence with ENSO BEST index. Moreover the magnitude of this anomaly is consistent between all these different satellites, in both spatial
pattern (first SV map) and temporal evolution (values of the temporal weights).

The location of this maximum/minimum in the Indonesian region where strong convection results in the greatest level of stratosphere/troposphere water vapour exchange. It is thus a particularly important phenomenon to be correctly represent in climate models and the SVD analysis presented here could provide a way to test this.

## 5   Conclusions

In this paper, a method for detecting and comparing the modes of variability in diverse datasets using singular value decomposition has been described. This analysis and the use of de-seasonalised monthly anomalies, which remove long-term trends and biases between datasets, has enabled a comparison of satellite data on cloud properties over three decades with the variation of a range of climate indices. The spatial pattern and timeseries of the leading SVs for three independent global data sets on both cloud fraction and cloud-top height, between latitudes of 60° S and 60° N has been computed. This analysis, derived from

**Figure 3.** Maps on the left side are the firsts four spatial SV for CTH from Cloud CCI; Plots on the right show the associate time series of the temporal weights in black line and the fitting with climate indices in red. The green line represent the offset and the slope obtained with the fitting, legend on top of each plots show the winning index together with lag and correlation coefficient (r).

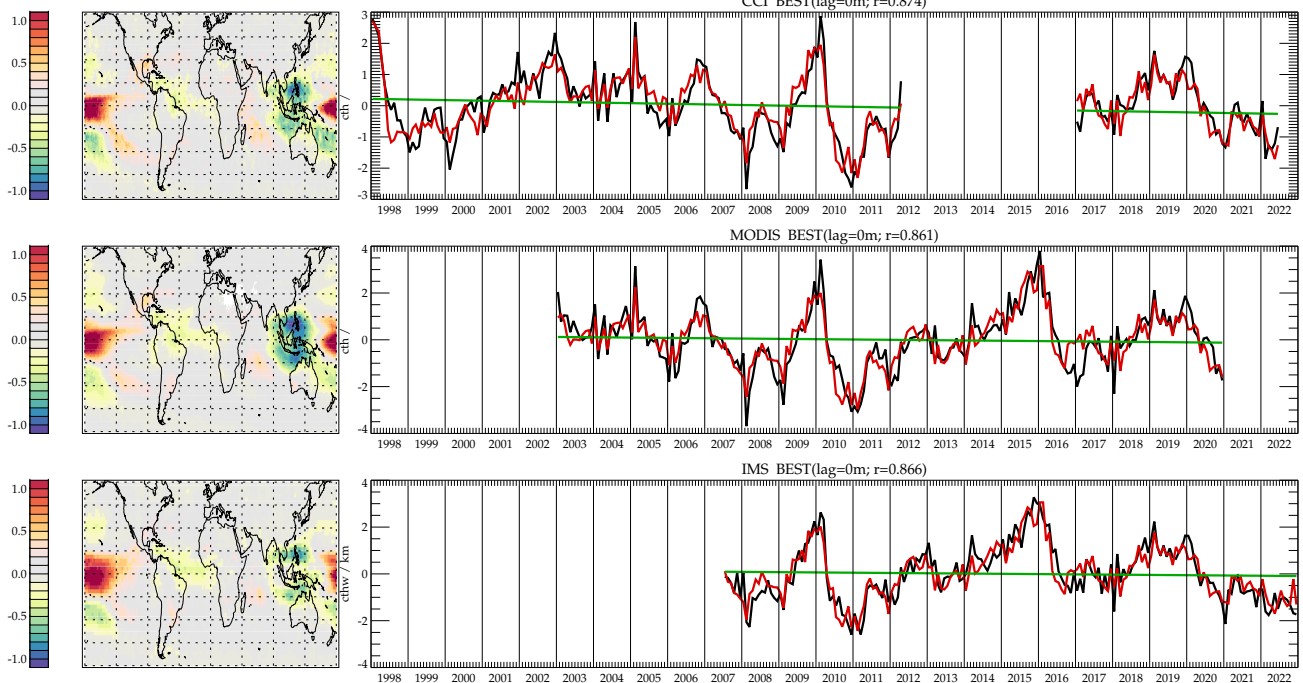

**Figure 4.** Fist mode and time series fitting for CTH from Cloud CCI, MODIS and IMS, following the colour scheme used in figure 3.

five polar orbiting satellites and covering different time periods over a 25 year time span, is shows to be strong and consistent correlation with ENSO indices.

Representation of clouds in climate and Earth system models is key, due to the importance of cloud feedback in the system. However, the complexity and spatial scales of processes governing cloud distributions and their temporal variability is challenging to accommodate. Furthermore, cloud variables in climate and Earth system models differ from those retrieved from satellite observations such as those used in this paper, so direct comparisons are not straightforward and involve complicated satellite simulators. However, they would be amenable to the singular vector decomposition method as applied to satellite data in this paper.

Comparison of model and satellite respective SVs, temporal weights time series and their fits to ENSO and other established climate indices could potentially offer a new approach to evaluate consistency between such models and existing satellite data sets, which are now multi-decadal and are to be extended through planned future satellite missions.

Furthermore, the analysis approach presented here could be applied to any long-term gridded dataset, enabling the comparison of, and illuminating connections between, seemingly unrelated data. The use of de-seasonalised anomalies in this analysis

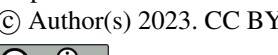

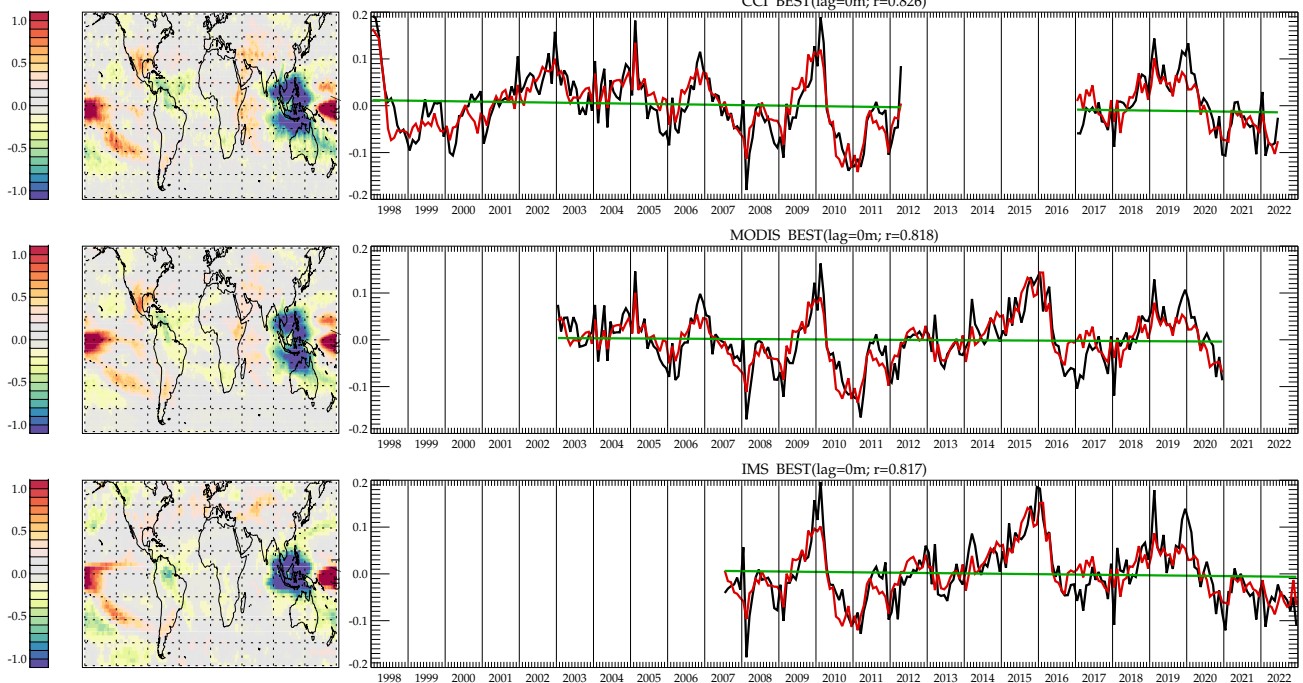

**Figure 5.** Fist mode and time series fitting for CFC from Cloud CCI, MODIS and IMS, following the colour scheme used in figure 3.

is also an important aspect, as it removes the both the dominant seasonal cycle and instrument/dataset specific biases from the
analysis, which would otherwise obscure the longer-term variability observed here.

*Data availability.*   Cloud CCI data, (A)ATSR L3C: https://climate.esa.int/en/projects/cloud/data/

C3S data, SLSTR-A Cloud Properties : https://cds.climate.copernicus.eu/cdsapp#!/dataset/satellite-cloud-properties?tab=overview

MODIS data: Platnick, S., et al., 2015. MODIS Atmosphere L3 Monthly Product. NASA MODIS Adaptive Processing System, Goddard
Space Flight Center, USA: http://dx.doi.org/10.5067/MODIS/MOD08_M3.061

IMS data: https://dx.doi.org/10.5285/489e9b2a0abd43a491d5afdd0d97c1a4

*Author contributions.*   E.C. perform the Cloud_cci and MODIS analysis and drafted the paper, G.T. contributed to the analysis and the writing
of the paper, R.S. developed the SVD analysis code and perform IMS analysis, B.K. oversight of research and contribution to drafting paper.
All authors contributed in developing of the method, discussing the results and revising the paper.



*Competing interests.* No competing interest.

*Acknowledgements.* Dr. Martin Stengel (DWD) for leadership of the ESA CCI Cloud project and Drs Simon Pinnock and Michael Eisinger (ESA) for technical management. UK Centre for Environmental Data Analysis (CEDA) for use of Jasmin compute infrastructure to process satellite data and for archival of ATSR-2, AATSR, Sentinels-3A -B, MODIS and IMS data used in this analysis. NERC National Centre for Earth Observation for co-funding the research.



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
