# Peer review of "Singular Vector Decomposition (SVD) of satellite datasets: relation between cloud properties and climate indices"

_Atmospheric Measurement Techniques, 2023_

## Author Comment (AC1)

We thank Referee 1 for giving careful attention to this paper.

In the following the referee comments are in 'black' and our answers are in 'blue'

In response to Ref 1's concern, to provide more background for the reader, we add here the following brief text in regard to SVD, EOF and PCA approaches. This can be added to the "Introduction" or in "Supplementary Material".

"Singular Value (or Vector) Decomposition (SVD)', 'Principal Components Analysis (PCA)' and 'Empirical Orthogonal Functions (EOF) are used interchangeably in the literature.

SVD is used in linear algebra to refer to the decomposition of an $m{\times}n$ matrix $\mathbf{A}$ of the form $\mathbf{A} = \mathbf{USV}^T$, where $\mathbf{U}$ and $\mathbf{V}$ are $m{\times}m$ and $m{\times}n$ matrices, and $\mathbf{S}$ is an $m{\times}m$ diagonal matrix. The columns of $\mathbf{U}$ and $\mathbf{V}$ are referred to as singular vectors), and form sets of orthornormal bases of $\mathbf{A}$. The diagonal elements of $\mathbf{S}$ are referred to as singular values, and essentially represent the weighting of each basis vector pair when reconstructing $\mathbf{A}$. In our case, $\mathbf{A}$ consists of spatial distributions of a geophysical parameter in one dimension and the other is the time dimension. SVD produces a set of orthogonal singular vectors representing the patterns of spatial variability and the set of amplitudes which constitute their time series. This set of singular vectors, alternatively named empirical orthogonal functions (EOFs), or principal components are determined directly from the data matrix.

The term principal components analysis (PCA) is more frequently used in computer science/image processing, as a dimensionality reduction technique. It refers to a compact representation of data that minimize the information loss. PCA uses SVD to find the singular vectors that reproduce most of the variance of the dataset. Rather than using all the singular vectors, the series is truncated to the leading n singular vectors or principal components to reconstruct the data set.

In environmental sciences SVD is used to decompose a data set into the spatial patterns, referred to as Empirical Orthogonal Functions, which explain the maximum amount of variance in a two-dimensional data set. One dimension in the data set represents the dimension in which we are seeking to find structure (usually spatial structures or lat/lon distributions, as in this paper), and the other dimension is that in which realizations of this structure are sampled (usually time, as in this paper). The set of structures in the sampling dimension (e.g. time) can be referred as the Principal Components (PC's), and they are related one-to-one to the EOF's. Both sets of structures, the one in spatial dimension (called SV in the paper) and the one in time dimension (called temporal weights in the paper) are orthogonal.

SVD can also be applied to the covariance matrix *between* two datasets. In this case, SVD picks out structures in one data set that are best correlated with structures in the other data set or vice versa. These are structures that 'explain' the maximum amount of the covariance between the two data sets, in a similar way that EOF's and PC's are the structures that explain the most variance in a single data set. It is reasonable to call this Maximum Covariance Analysis (MCA).

In the table below we report the different terms that have been used in different fields/literature. First row refer to the terms used in this paper and follow the decomposition of the dataset matrix with SVD: $\mathbf{A} = \mathbf{USV}^T$

| Analysis | v | Sii | Uij |
|---|---|---|---|
| Singular Vector Decomposition (SVD) | Singular vector | Singular value | Weights (time-series of temporal weights) |
| EOF analysis | EOF | Singular value | Principal Components or Amplitude time-series |
| Principal Component Analysis (PCA) | Principal Component or loading pattern | | Expansion coefficients or Expansion coefficient time-series |
| Eigenvector analysis (this is only for square matrix, often used with covariance matrix) | Eigenvector | Eigenvalue | Eigenvector |

This scientific paper presents the application of a statistical (or algebraic) technique that identifies that portion of data in a time series that is useful in interpreting the dynamics of the atmospheric system through correlations with climate indices. Specifically, the technique presented is one of the possible belonging to the group of the Eigen techniques, the Singular Value Decomposition (SVD), and the time series are those of cloud properties derived from satellite measurements.

After reading the manuscript, I find myself in the uncomfortable position of rejecting the work and having to suggest that the authors rethink the logic of the study from the ground up. This work is not only incomplete, it is also wrong in its logical premises, the type of data used, the technique presented and fail novelty. I set out my points below.

1) Is this scientific journal the most appropriate for such work? I don't think so. This manuscript presents no new algorithm, no validation of the data, no error analysis of the data. In truth, there is only the application of a decomposition technique of a historical series and correlations with climate indices.

   The authors themselves indicate in the abstract as an application to assess the performance of global climate models. AMT is not the appropriate venue for this type of message. Perhaps ACP (I have my doubts), but even better would be ClimDyn or JCLIM.

A previous paper (Li et al 2020) presents a similar analysis of total cloud cover and superimposes three different climate indices associated with El Nino and SST anomalies, to study the relation between cloud cover and sea surface temperature. Li et al (2020) also take the further step of rotating the first two singular vectors to minimise the correlation between them, in an effort minimise "mode mixing", whereby multiple single vectors are driven by the same underlying process. The novel aspects of our paper compared to Li et al (2020) are:
   1) Our analysis allows for a temporal offset and linear trend between the climate index and temporal weights of the cloud property singular vectors.
   2) Rather than rotating singular vectors, our analysis allows for a linear combination of different climate indices to be fitted to each singular vector, while providing a correlation between each index and the SVD temporal weights.
   3) We apply the SVD analysis to both cloud cover and cloud-top height.
   4) We apply the same analysis to three different and wholly independent datasets, with one (the IMS data) being very different in measurement principle, spatial resolution and sensitivity to cloud properties, and find very consistent results from all three.

Thus, we would assert that there is significant novel and original content in our work to warrant publication. Furthermore, we would counter the suggestion by the reviewer that this work is not suitable for publication in Atmospheric Measurement Techniques, by suggesting that this analysis is, in fact, novel in the atmospheric measurement and remote-sensing communities. This is evidenced by the fact that similar analyses by the climate modelling community only became known to us once we undertook the task of writing up our own analysis. We also feel that there can be no denying that the presented analysis is well aligned with the stated purpose of Atmospheric Measurement Techniques: "The main subject areas comprise the development, intercomparison, and validation of measurement instruments and techniques of data processing and information retrieval for gases, aerosols, and clouds."

As an aside, we have added references in the paper to technical documentation detailing the validation of the data used in this paper.

- Product Validation and Intercomparison Report (PVIR), v6.1. ESA Cloud_cci. https://climate.esa.int/media/documents/Cloud_Product-Validation-and-Intercomparison-Report-PVIR_v6.0.pdf. Last accessed on 23/05/2024
- Algorithm Theoretical Basis Document, v.6.2. ESA Cloud_cci. https://climate.esa.int/media/documents/Cloud_Algorithm-Theoretical-Baseline-Document-ATBD_v6.2.pdf Last accessed on 23/05/2024

2) With reference to the technique presented (SVD), the choice is unsatisfactory. A characteristic hurdle in climate research is the size of the phase space. In a real system such as the climate it is basically infinite and in GCM is quasi-infinite. Clearly, the observations are per se limited and there is the need to single out those significant first-order components that best describe the underlying dynamics of the system. In the pursue of which, the direction in the phase space assumes a relevant role (which by the way justifies the difference between statistics and the isotropic statistical mechanics). Long story short: to achieve this goal we have already a myriad of Eigen Techniques, better suited to this purpose. Alone the EOFs, the rotated EOFs, or climate networks (e.g. Ludescher et al 2014, Donges et al. 2011, 2015). There are more out there that are more sophisticated and appropriate approaches. Why is this the case? Because in all the techniques I have mentioned it is possible to embed constraints to analyze what really counts: the variability. The argument that SVD is preferable to other techniques because it is simpler is true at the expense of depth and accuracy of analysis. With all due respect to colleagues, this strikes me as more of a task for a freshman in a master of science course than for established researchers aiming at novel results. Moreover, nowadays, thanks to the open science paradigm, there are many public repositories where implementations of the respective techniques can be conveniently downloaded. Thus, the objection of having to code everything from scratch no longer exists.

As explained above, EOF and SVD analysis are simply different names for the same procedure. The reviewer also seems to equate a more complex analysis with a better analysis, which is certainly not always the case. We are of course aware of publicly available tools for statistical analysis and have made use of some in the analysis presented in this work (in particular the IDL interface to LAPACK). In this paper we have applied an analysis technique to three independent data sets and are reporting on the results in order to illustrate the potential value of these three data sets and the applied technique. Widespread use and further exploration of the data with techniques other than those used in this study would be welcome, but would not negate the work as presented here.

The reviewer raises a valid point that a more quantitative description of the variability shown in each singular vector could easily be included. This has been done through the addition of the following statistics to the figures in the paper:

1) The percentage of the variability of the data described by each singular vector, where the variability is the variance of either the cloud cover or cloud-top height anomaly about the monthly-mean for each dataset).

2) The standard deviation of the temporal weights of each singular vector, which gives a representation of the typical scale of the anomalies in cloud-top height or cloud-cover.

3) The choice of data. The authors cannot use L3 monthly averages for their analysis, rather they must use L2 in conjunction with the standard deviation (or variance) and respective error metrics and ask how the aggregation of the L2 time series impacts the goodness of fit. This is because each sensor has a different spatio-temporal sampling and the patterns emerging from the analysis are affected by these differences.

In other words, the authors must first try to answer the question of what correlation length is required for different sensors to represent the same cloud field. No data set can ever represent reality. So every data set is exposed to the same shortcomings. Before looking for a quasi-orthogonal basis to link to a climate index, one must make the observational data sets as homogeneous as possible among each other. And introduce error metrics.

In our view, this comment suggests an incomplete understanding of satellite data products on the part of Ref 1. The use of monthly data (Level-3) for our analysis rather than individual soundings (Level-2) is not only valid, but is a key feature. One important point is that the data used in this paper were all derived from sun-synchronous polar orbiting sensors with morning equator crossing times between 9 and 10 am, which thereby restricts the range of diurnal variability between them. Then, when looking at monthly aggregated data, we are not comparing data of specific clouds, but the average cloud properties seen in each grid box over a month. It would be our contention that over a month the satellite observations provide a dense enough sampling that the average (and standard deviation etc) is robust against the differences in sampling between the three instruments. Furthermore, the use of monthly aggregated data ensures that the data being fed into the SVD analysis are spatially complete, with valid data in each grid cell for each month, which is a prerequisite for the matrix decomposition to work.

We would strongly dispute Ref 1's statement that "no measurement can represent reality" (if that were the case, there would be no point in making the measurement), but it is true that every measurement is subject to the limitations of its own sampling, sensitivity and error characteristics. The representativeness of the monthly means used here is demonstrated by the consistency of the independent CCI and MODIS datasets (see Fig.1), which have similar sampling and sensitivity. This is clearly not so for the IMS data, which represent the same distribution of clouds and their variability, but with a very different sensitivity to the cloud properties themselves (due to the very different measurement technique).

Comparing instantaneous observations of individual clouds, or "cloud fields" (the definition of which is not clear) is obviously a valid approach to comparing data from different satellites (provided they make measurements which are close enough in temporally and spatially, as the reviewer suggests), but such comparisons aren't going to be directly relatable to climate indices or multi-annual variability. Furthermore, they are even more fraught with difficulty than the reviewer appears to realise, as differences in viewing geometry, illumination, sensitivity and pixel size can all confound any comparison of complex, 3-dimensional structures like clouds.

4) The manuscript is not consistently elaborated because all climate indices are stated in a table (superfluous at this point), but only ENSO is mentioned in the manuscript. From the title chosen by the authors, the manuscript suggests a (laudable and ambitious) generalisation, but this is nowhere to be found.

The table lists all the climate indices which have been fitted to the SVD temporal weights time-series, so it is not superfluous. In the end it, perhaps unsurprisingly, turns out that El Nino dominates the cloud-top height and cloud cover between -60° and 60° latitude. By limiting the analysis to different regions, the signals of other indices do become more prominent, as is shown for the Northern Hemisphere Atlantic in the figures shown below, where the Arctic Oscillation is prominent. However, we did not want to go down the path of providing a myriad of SVD analyses for different regions picking out different climate indices. We do take the point that the full table should perhaps be moved to an appendix or supplementary material, rather than appearing in the body of the paper. The same could be said for some example regional SVDs, showing correlations with different climate indices.

[Figure]

In this figure maps on the left side are the firsts four spatial SVs of cloud fraction (CFC) from IMS dataset; the legend above each map presents: the rank of singular vector (SV1, SV2…), the standard deviation of the temporal weights and the percent of variance associated with the singular vector. Plots on the right show the associated time series of the temporal weights (black lines) and the fit with climate indices (red lines). The green lines represent the offset and the slope obtained in the fitting. The legend on top of each plot shows the leading index together with the lag, correlation coefficient (r), the significance and number of standard deviations by which the distributions (of square differences between black line and red line) deviates from its null-hypothesis expected value (independent, uncorrelated random numbers)

5) Figures 2 and 3. I invite the authors to fill in the time gap between 2012 and 2017. Without coverage of these dates, it is not even remotely conceivable for me to scrutinise the results as the trend and eventual statistical significance (which is missing by the way). There are no prerequisites because the data presented are lacking.

The paper makes it clear that the data has been detrended and is only concerned with the spatial patterns of variability with the data. We acknowledge that differences in sensitivity and potential calibration issues between the different sensors mean that looking at the time series of the full CCI dataset (and potentially the other two data records as well) is likely to produce spurious trends and step-changes, so we have avoided this. There is no trend to scrutinise.
It would be nice to have continuous coverage between the AATSR instrument and its successor the Sentinel SLSTR, but the fact remains that Envisat failed in 2012, five years before SLSTR data became available. Using a 3rd party instrument to fill this gap in the CCI data record, applying the same cloud retrieval scheme, would offer a potential basis for a future study.
Furthermore, the inclusion of two independent data records, which fill this gap, while not providing coverage at the start and/or end of the whole time-period, means that we do present full temporal coverage. And, finally, the consistency of the results between all three data records demonstrates that the analysis is robust with respect to this data-gap.
We do acknowledge that the statistical significance of our comparisons had not been indicated, however, these have now been included with the plots.

6) As a final point, I suggest that the authors, for the next draft of the paper, reserve some of their time and effort for interpreting the results they find. The correlation of CTH with ENSO, for example, is straightforward and immediately understandable on the basis of basic arguments of fluid- and thermo-dynamics. The matter becomes more interesting if one creates Hovmoller graphs of correlations between certain cloud properties and climate indices. The evolution of these teleconnections may reveal as yet unknown aspects. For example, the timing of the start of the monsoon season or exchanges of energy and momentum between low latitudes and the poles. Be that as it may, it is not an easy task, precisely because the authors want to tackle it empirically, on the basis of data and not models. But precisely for this reason, from my point of view, the highest possible precision must be requested in the formulation of the problem.

The purpose of this paper was to introduce and demonstrate the use of SVD (or EOF, PCA etc) in the analysis and comparison of satellite data to the atmospheric measurement / satellite remotes sensing community, where it is not widely known. Thus, we feel extending the analysis is beyond the scope of the current paper. It is intended to extend the scope of this initial analysis in future papers for publication in atmospheric science or climate journals and involving collaboration with climate modelling experts.

---

## Author Comment (AC2)

We thank Referee 2 for the time spent in this revision, the suggestions and feedbacks.

In the following our answers are in 'blue' and the referee comments in 'black'

**General Comments:**

This paper demonstrates an application of the Singular Value Decomposition (SVD) statistical technique to correlate cloud properties observed in satellite data against climate indices.

SVD is commonly-used in the atmospheric sciences as a method to understand the spatio-temporal variability of geophysical data. The authors effectively demonstrate how one can use SVD to compare the modes of variability in a long observational satellite record against climate indices.

Because the authors both a. emphasize the applicability of the technique to any gridded satellite dataset (not just atmospheric fields), and b. position this methodology as primarily novel for climate model validation, this paper may be better suited for a journal focused more generally on climate rather than atmospheric measurements.

While SVD is an established methodology in atmospheric sciences and its novel application here to multiple satellite time-series serves to indicate future potential of this approach for comparison of observations with climate models, the paper does not report new findings on climate per se. We therefore consider AMT to be appropriate rather than climate journals.

That said, I think it would be appropriate for publication in Atmospheric Measurement Techniques with revisions addressing the specific comments below.

**Specific Comments:**

1. The study is framed as a technique that can be effectively generalized to any gridded dataset, but only provides examples of cloud observations. An additional example would be helpful to support the generalization.

   We have also applied the SVD technique described in the paper to a number of other atmospheric parameters in the IMS dataset.

   We illustrate here one such example for stratospheric ozone (between 24-27km) where the temporal weights, associate with the first SV, are fitted with QBO index. Analyses of atmospheric parameters other than cloud are intended in future publications.

[Figure]

Maps on the left side are the first four spatial SVs for ozone between 24-27km from IMS the dataset; the legend above each map present: the number of singular vector (SV1), the standard deviation of the temporal weights, the percent of variance associate with the singular vector. Plots on the right show the associated time series of the temporal weights (black line) and the fits with climate indices in red, while the green lines represents the offset and the slope obtained in the fitting. The legend above each plot shows the best fitting climate index together with lag, correlation coefficient (r), the significance and number of standard deviations by which the distribution deviates from the from the null-hypothesis.

2.  The authors report strong correlations in the results but omit statistical significance.

Statistical significances 'sign' are now added in the time series plots titles, together with 'nstd' the number of standard deviations from the expected value from the null-hypothesis . Both values are obtained with the IDL r_correlate routine (*Numerical Recipes, The Art of Scientific Computing (Second Edition)*, Cambridge University Press (ISBN 0-521-43108-5).

[Figure]

These plots are the new versions of figs 4 and 5 of the paper, with the addition of significances (sign) and numbers of standard deviations (nstd) from the expected value from the null-hypothesis in the labels together with lag and correlation.

For CFC and CTH the significance is less than 1.5x $10^{-40}$ and 'nstd' is greater than 10 for all the 3 datasets.

3. Spend time exploring what the authors themselves state as one of the novel aspects of using this technique: understanding the underlying causes of the variability.

The intention of the paper is to describe the methodology and point towards its potential for such future analyses. Although we agree with the Referee that inclusion would

strengthen the paper, extensive further work would be entailed to incorporate climate modelling which we view to be out of scope and not strictly necessary for the paper to serve its intended purpose.

4. Did the authors try any other decomposition methods? How does using SVD compare to other methods such as using EOFs or other approaches? A clear statement for why this technique was chosen and what its limitations are should be included.

SVD is the procedure used to derive temporal weights for spatial EOFs through the decomposition of a matrix. There is a lot of different notation used in the literature to describe what, in the end, provide very similar ways for decomposing a matrix into orthonormal basis vectors. Please see also the introduction to our answer Referee 1.

We decompose our matrices in space and time patterns that represent the most variance of the dataset. We can add a paragraph explaining this in the paper.

One additional step in this analysis which we may investigate in the future is the rotation of singular vectors in order to minimise the correlation between the different principle basis vectors, but this has not been attempted in the presented analysis and is beyond the scope of this paper.

**Technical Corrections:**

- Paragraph 45: typo: de-seansonalised
- Paragraph 145: typo: de-sesonalized
- Paragraph 185: type: series off time-lags
- Figs 3, 4, 5: consider centering the maps on the Pacific Ocean instead of the Atlantic to better show the ENSO correlation

Maps are now centred in the Pacific. Our thanks to Ref 2 for these suggestions!